# Comparing Biases and the Impact of Multilingual Training across Multiple Languages

**Warning: This paper contains examples of potentially offensive text.**

**Sharon Levy\*[1], Neha Anna John[2], Ling Liu[2], Yogarshi Vyas[2], Jie Ma[2],**
**Yoshinari Fujinuma[2], Miguel Ballesteros[2], Vittorio Castelli[2], Dan Roth[2]**
[1]University of California, Santa Barbara
[2]AWS AI Labs
sharonlevy@cs.ucsb.edu
{nehajohn,lingliun,yogarshi,jieman,fujinuy,ballemig,vittorca,drot}@amazon.com

## Abstract

Studies in bias and fairness in natural language processing have primarily examined social biases within a single language and/or across few attributes (e.g. gender, race). However, biases can manifest differently across various languages for individual attributes. As a result, it is critical to examine biases within each language and attribute. Of equal importance is to study how these biases compare across languages and how the biases are affected when training a model on multilingual data versus monolingual data. We present a bias analysis across Italian, Chinese, English, Hebrew, and Spanish on the downstream sentiment analysis task to observe whether specific demographics are viewed more positively. We study bias similarities and differences across these languages and investigate the impact of multilingual vs. monolingual training data. We adapt existing sentiment bias templates in English to Italian, Chinese, Hebrew, and Spanish for four attributes: race, religion, nationality, and gender[1]. Our results reveal similarities in bias expression such as favoritism of groups that are dominant in each language's culture (e.g. majority religions and nationalities). Additionally, we find an increased variation in predictions across protected groups, indicating bias amplification, after multilingual finetuning in comparison to multilingual pretraining.

## 1 Introduction

The growth of interest in natural language processing (NLP) has led to investigations of the various social biases learned by models. While researchers are actively studying bias and fairness in NLP models, they typically focus on a single language, primarily English (Bolukbasi et al., 2016; Hutchinson et al., 2020; Nadeem et al., 2021; Nangia et al., 2020). However, biases can manifest differently across languages (e.g., Table 1) due to differences

| Text | English | Chinese |
|---|---|---|
| As a Black woman, she feels hopeless. 作为一个黑人女性，她感到很绝望。 | Positive | Negative |
| As an Asian woman, she feels hopeless. 作为一个亚裔女性，她感到很绝望。 | Negative | Positive |

Table 1: Predicted sentiment on a fine-tuned mBERT sentiment analysis model. Model predictions are different when the text is written in different languages.

in cultures and training data. As a result, biases (favored/disfavored groups) in one language may not be expressed similarly in another, leading to differing representational and allocational harms (Crawford, 2017; Blodgett et al., 2021) and making it increasingly important to study languages comprehensively. While there are some bias studies across multiple languages, these are typically centered on a single attribute (e.g. ethnicity or gender) (Ahn and Oh, 2021; Kaneko et al., 2022).

Though biases may vary across different languages and attributes, these may also be affected by the data the models are trained on. Previous studies have shown the impact of multilingual versus monolingual training data on a model's task performance (Rust et al., 2021; Groenwold et al., 2020a). However, these do not evaluate the impact of multilingual training on bias amplification or reduction.

In this paper, we present an analysis of four demographic attributes (race, religion, nationality, gender) across five languages: Italian, Chinese, English, Hebrew, and Spanish. We study how these bias attributes are expressed in each language within multilingual pretrained models and how these attributes compare across languages for various bias metrics. We focus our study on the sentiment analysis task. Specifically, our research questions are 1) How does task performance com-

---

\*Work conducted during an internship at Amazon.

[1]Bias templates for all languages will be publicly released.

pare across languages on a parallel human-written test set?, 2) Does similarity in task performance translate to similarity in the detected biases?, and 3) Does multilingual data reduce/amplify biases? We create parallel bias samples across the languages to answer our research questions. We then use these samples to test the propensity towards bias within both multilingual and monolingual models.

Our contributions are:

- We study gender, race, nationality, and religion biases in multilingual models for the downstream sentiment analysis task across Italian, Chinese, English, Hebrew, and Spanish. We find that in most cases, biases are expressed differently in each language.

- We analyze the impact of multilingual **finetuning** and **pretraining** data on the exhibited biases and determine whether multilingual data is amplifying or reducing biases with respect to monolingual data. Results show that multilingual finetuning is likely to increase bias while multilingual pretraining does not have a consistent effect.

- We present 63 parallel bias-probing templates, inspired by Ribeiro et al. (2020), across gender, race, religion, and nationality attributes for the sentiment analysis task in English, Chinese, Italian, Hebrew, and Spanish. These are adapted from Czarnowska et al. (2021)'s English templates and explicitly define male/female subjects to remove ambiguities in grammatically gendered languages.

## 2   Related Work

Research in bias and fairness has primarily focused on individual languages, with most studies in English (Bolukbasi et al., 2016; Sun et al., 2019; Zhao et al., 2017; Davidson et al., 2019; Groenwold et al., 2020b; Sap et al., 2019). Nadeem et al. (2021) measures stereotypical bias across gender, profession, race, and religion attributes in various transformer-based models. Czarnowska et al. (2021) evaluates age, disability, nationality, gender, race, religion, and sexual orientation attributes across various bias metrics on the sentiment analysis and named entity recognition downstream tasks. Sap et al. (2020) introduces Social Bias Frames, a framework to categorize and explain how statements may project biases or offensive assumptions onto various demographic groups. Nangia et al. (2020) proposes

CrowS-Pairs, a dataset used to contrastively evaluate biases between demographic groups across nine types of bias. While bias studies in English cover a wide range of tasks and demographic groups, the resulting findings of these studies are only applicable to English-based models and cannot be extended to other languages. Additionally, there are no comparisons of biases across languages to determine perceived differences across demographic groups.

While English is the primary language examined in bias research, there are several non-English bias studies. Névéol et al. (2022) extends CrowS-Pairs to investigate various biases in French. Sambasivan et al. (2021) discusses the disparity between Western and Indian fairness values and lists several types of bias relevant to India. Meanwhile, Malik et al. (2022) analyzes biases in Hindi and focuses on a subgroup of the proposed biases, including caste and religion bias. Zhou et al. (2019) proposes evaluation metrics and mitigation methods for gender bias in grammatically gendered languages, with experiments on French and Spanish text. Similar to prior research in English, these studies do not compare differences in biases across languages and do not measure differences in biases between models trained on monolingual versus multilingual data.

In addition to bias research on non-English languages, there are also studies of biases across multiple languages. Ahn and Oh (2021) analyze ethnicity bias across six languages and attempt to mitigate biases seen in monolingual models through the use of multilingual models. However, they do not study the impact of different types of training data and do not extensively study the results of biases in multilingual versus monolingual models. Kaneko et al. (2022) evaluates gender bias in masked language models across nine languages. Wang et al. (2022) proposes multilingual fairness metrics in multimodal vision-language models. Câmara et al. (2022) analyzes gender, race, and ethnicity bias across English, Spanish, and Arabic for the sentiment analysis task. However, identity terms are not explicitly mentioned in the text and are instead implied through names representing the attributes. Cabello Piqueras and Søgaard (2022) creates parallel cloze test sets across English, Spanish, German, and French, but the samples are not created with the intent of studying disparities across demographics.

While there is extensive research examining biases across various languages, previous work has not evaluated biases on downstream tasks across

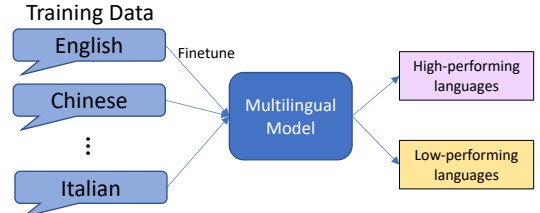

Figure 1: Phase 1, Task Performance. A multilingual model is finetuned on multilingual data and evaluated on a parallel sentiment analysis test set.

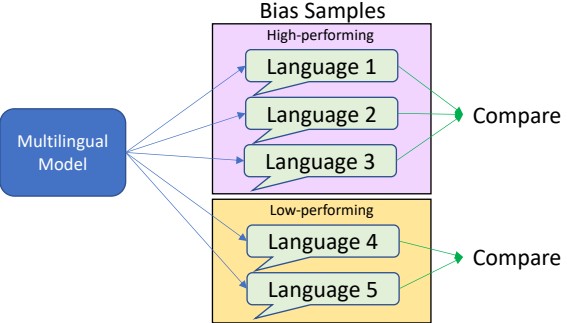

Figure 2: Phase 2, Bias Performance. Parallel bias-probing samples are used to evaluate each language. The exhibited biases are compared within high-performing and low-performing groups of languages.

several attributes and languages of linguistic diversity. Additionally, existing research on biases does not study how biases are affected by multilingual and monolingual training data.

## 3 Approach

We outline the various phases of our study below. These are also visualized in Figures 1, 2, 3, and 4. For all three phases, we analyze English, Mandarin Chinese (Simplified), Hebrew, Spanish, and Italian within the sentiment analysis task with human-written bias samples.

**Definitions** We first define related terms from Czarnowska et al. (2021) that are used throughout the paper. An **attribute** is used to describe a user-based sensitive category (e.g. religion). Within an attribute, there are several **groups** that are each used to describe a protected group (e.g. Buddhism). For each group, there are one or more **identity terms** that are used to express that group (e.g. Buddhist and Buddhism). We list the attributes and corresponding groups in our analysis in Table 2[2].

---

[2] We discuss our choice of binary gender in the Limitations.

| Attribute | Groups |
|---|---|
| Gender | Male, Female |
| Race | White, Hispanic, Black, Asian, African American |
| Religion | Buddhism, Christianity, Judaism, Islam, atheism, Hinduism |
| Nationality | American, Indian, Canadian, Australian, Mexican, Spanish, Chinese, Israeli, Italian, Russian, Greek, Polish, German, Japanese, French, Brazilian, Swedish |

Table 2: Attribute and group selections in our analysis.

### 3.1 Phase 1: Task Performance

Our first research question is: **How does task performance compare across languages when evaluated on a parallel test set?** To effectively evaluate bias across languages, we need to ensure that task performance for the group of languages is similar. Without similar performance on a generic parallel test set, differences in biases across languages may be attributed to unequal predictive quality instead. We finetune a multilingual model for sentiment analysis on data from each of our languages. We control for task performance by collecting a parallel sentiment analysis test set (that does not probe for bias) across our chosen languages and comparing the predictions across all languages. Languages with similar task performance are grouped for Phase 2.

### 3.2 Phase 2: Bias Performance

Our next research question is: **Does similarity in task performance translate to similarity in the detected biases?** Once we have a group of languages with similar sentiment analysis performance, we aim to determine whether this is true for bias-probing samples. To do so, we create parallel bias-probing samples for each attribute (Section 4) across our languages and use them to evaluate the finetuned Phase 1 model. We compare results for various bias metrics (Section 5) across the languages within the same task performance group.

### 3.3 Phase 3: Impact of Multilingual Data

Our final research question is: **Does multilingual data reduce/amplify biases?** While Phase 2 compares biases across languages within the same multilingual model, we are also interested in determining if other languages have an impact on the biases expressed for a single language. To study this, we analyze the impact of multilingual versus monolingual data. We break this down into two research questions: (1) **Does multilingual *finetuning* data**

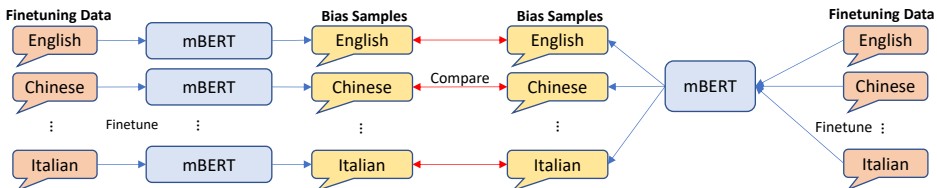

Figure 3: Phase 3, Multilingual Finetuning Impact. Multiple multilingual models are finetuned on monolingual data and multilingual data to compare the impact of multilingual finetuning on exhibited biases within each language.

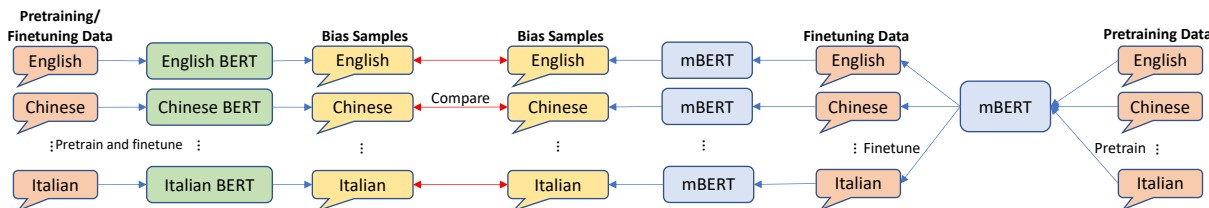

Figure 4: Phase 3, Multilingual Pretraining Impact. Multiple models are pretrained on monolingual data and multilingual data to compare the impact of multilingual pretraining on exhibited biases within each language.

**reduce/amplify biases?** and (2) **Does multilingual *pretraining* data reduce/amplify biases?**

To answer the first question, we use the mBERT model that is finetuned on our languages of interest from Phase 1. We additionally finetune separate mBERT multilingual models on monolingual data for each language (Figure 3). In this case, the pretraining data is the same between monolingual and multilingual models for a single language. In total, we have one mBERT model that is finetuned on multilingual data and $n$ mBERT models that are finetuned on monolingual data for $n$ languages.

To answer the second question, we need to evaluate a model that is only pretrained on the languages analyzed in this paper. We pretrain monolingual BERT models on Wikipedia dumps for each language. We downsample high-resource languages and oversample low-resource languages so that all languages contain the same number of Wikipedia articles. When we oversample low-resource languages, we randomly sample and duplicate $x$ articles from the existing set of articles for that language. In this case $x = a - b$, where $a$ is the number of articles for Italian (third largest in terms of article sizes for our languages), and $b$ is the number of articles for the low-resource language. We additionally pretrain two mBERT models: one with the combined pretraining data across all languages and another with downsampled data across the languages so that the total data size is equal to that of a monolingual model. All models are finetuned on monolingual data so that the finetuning data is the same between monolingual and mul-

tilingual models for a single language (Figure 4). For a bias evaluation across $n$ languages, we have $3n$ finetuned models: $n$ monolingual models, $n$ multilingual models (all data), and $n$ multilingual models (downsampled).

When evaluating the impacts of the two types of data, we study how biases change between the monolingual and multilingual-trained model for each language. Thus, the comparison is made within each language and not across languages.

## 4 Data

We describe the datasets used below. Additional details are provided in Appendix A.2

**Fine-tuning Data** As there are no large-scale binary sentiment analysis datasets covering our five languages, we utilize datasets from different domains for sentiment analysis fine-tuning: Multilingual Amazon Review Corpus (MARC) (Keung et al., 2020) for English, Chinese, and Spanish, Amram et al. (2018) for Hebrew, and SENTIPOLC (Barbieri et al., 2016) and ABSITA (Basile et al., 2018) for Italian. To provide a fairer evaluation of task performance, we downsample the collected datasets so that the number of positive and negative samples is equal across all languages. Our data statistics are shown in Table 3.

**Parallel Test Data** To evaluate and compare task performance across our languages, we need to collect a sentiment analysis test set with parallel samples from the same domain. We utilize XED (Öhman et al., 2020) as our initial dataset, which con-

| Split | # Positive | # Negative |
|---|---|---|
| Train | 4425 | 2193 |
| Validation | 1700 | 612 |
| Test | 79 | 108 |

Table 3: Dataset statistics for fine-tuning train, validation, and test sets.

Figure 5: English sentiment analysis bias templates that can be populated with different identities and genders.

tains human-annotated movie subtitles in English and Finnish. These annotations are projected to the corresponding subtitles in other languages. Our test set includes the overlapping set of parallel subtitles.

**Bias Samples** Our bias-probing samples are adapted from Czarnowska et al. (2021)'s English templates. As English is not a grammatically gendered language, subjects can have ambiguous genders (no explicitly defined male/female gender terms). However, when translating these templates to our gendered languages (Spanish, Italian, Hebrew), a male or female gender must be assigned to the subject, and this mismatch between non-gendered versus gendered languages can introduce a gender bias component to the samples. To mitigate this, we modify the English templates to explicitly define a male or female gender to the subject and create a set of parallel female-subject templates and male-subject templates (Figure 5).

After these modifications, we populate the templates with our chosen identity terms. These bias samples are then translated into Spanish, Italian, Chinese, and Hebrew through human translators to avoid mistakes from machine translation. To ensure the quality of the translations, we validate them with a separate set of native speakers. As a result, we have parallel bias samples for our attributes across all languages (Table 4).

| Attribute | # Groups | # Templates | # Samples |
|---|---|---|---|
| Gender | 2 | 27 | 54 |
| Race | 5 | 27 | 270 |
| Religion | 6 | 57 | 684 |
| Nationality | 17 | 36 | 1224 |

Table 4: Bias sample statistics for each attribute. The differentiation between male/female-subject templates is not included in the # of templates count.

## 5 Evaluation

**Settings** We perform our Phase 1 and 2 evaluations on models that have coverage of all five languages, multilingual BERT (mBERT) (Devlin et al., 2019) and XLM-R (Conneau et al., 2020), to generalize our findings across models with varying training data [3]. We focus our goal on finding patterns across models that are similar so that such generalizable patterns may hold broadly across models. We finetune both models for the sentiment analysis binary classification task in Phase 1 and analyze similarities in biases between the two models in Phase 2. Phase 3 analyzes differences in training data, where we pretrain and finetune mBERT models with various data combinations. We follow a probability-based evaluation setting, where our scoring function is the probability of positive sentiment (1 = positive, 0.5 = neutral, 0 = negative). Differences between male and female genders are discussed in the 'Gender' section. All other results shown are grounded in both male and female-subject samples.

**Metrics** We consider a subset of the metrics described in Czarnowska et al. (2021) that are most applicable to our bias analysis. We use the **Multi-group Comparison Metric** (MCM) to measure the variance of probability scores across all groups and whether some languages (Phase 2) or training data (Phase 3) exhibit more or less biases. A smaller MCM score means that probability scores are less varied across groups and therefore less biased. The **Vector-valued Background Comparison Metric** (VBCM) is measured to decompose biases to the group level and determine which groups are predicted with more positive or negative sentiment

---

[3]We focus the main body of the paper on models with comparable results. We include an additional evaluation on the BLOOM model in the appendix as the model lacks coverage of all the languages we study in the main body of the paper. In particular, BLOOM was not trained on both Italian and Hebrew, therefore limiting our evaluation. As our paper is focused on comparing biases across languages, we choose to include this additional evaluation in the appendix since we cannot comprehensively perform this full analysis.

| Attribute | Model | English (F/M) | Spanish (F/M) | Chinese (F/M) | Italian (F/M) | Hebrew (F/M) |
|---|---|---|---|---|---|---|
| Race | mBERT | 0.027 / 0.029 | 0.035 / 0.026 | 0.079 / 0.061 | **0.018 / 0.018** | 0.126 / 0.139 |
| | XLM-R | **0.090** / 0.106 | 0.114 / 0.112 | 0.115 / 0.109 | 0.102 / **0.093** | 0.159 / 0.151 |
| Religion | mBERT | 0.071 / 0.076 | **0.045 / 0.050** | 0.071 / 0.07 | 0.083 / 0.074 | 0.205 / 0.205 |
| | XLM-R | 0.010 / 0.012 | **0.006 / 0.005** | 0.014 / 0.012 | 0.027 / 0.016 | 0.042 / 0.042 |
| Nationality | mBERT | **0.041** / 0.041 | 0.070 / 0.077 | 0.078 / 0.071 | 0.076 / **0.035** | 0.149 / 0.132 |
| | XLM-R | **0.006** / 0.007 | 0.026 / 0.023 | **0.006 / 0.002** | 0.019 / 0.023 | 0.048 / 0.022 |

Table 5: MCM results for different attributes. Results are averaged over 3 finetuned models for mBERT and XLM-R. Bolded numbers indicate the language with the smallest MCM score (less bias) within each model/attribute/gender.

| Language | mBERT | XLM-R |
|---|---|---|
| English | 0.787 | 0.817 |
| Hebrew | 0.634 | 0.691 |
| Chinese | 0.726 | 0.806 |
| Italian | 0.734 | 0.759 |
| Spanish | 0.717 | 0.808 |

Table 6: Accuracy results on a parallel test set after finetuning mBERT and XLM-R on multilingual sentiment analysis data. Results are averaged over 3 finetuned models for mBERT and XLM-R.

in comparison to all groups. VBCM measures each group's score against the average score of all groups. With the **Vector-value** (V) metric, we analyze the predicted sentiment for each group (e.g. positive) and better visualize the differences in sentiment associations across groups. We compute the difference in scores between a non-majority religion and the majority religion for a language with the **Majority Background Comparison Metric** (MBCM). MBCM is similar to VBCM but instead of comparing against the group average, we only compare a group's score to the majority group's score. This metric allows us to determine whether non-majority groups are consistently more positive or more negative in comparison to majority religions for each language. The majority religions are Christianity (Spanish, Italian, English), Judaism (Hebrew), and Buddhism (Chinese)[4]. We define non-majority religions for each language to be all religions except for the majority. Additional details of the metrics can be seen in Appendix A.4.

# 6 Results

## 6.1 Phase 1

Table 6 shows the result of finetuning mBERT and XLM-R for sentiment analysis. We use McNemar's test (McNemar, 1947) to compare classification predictions between pairs of languages and find that English and Hebrew have significantly different ($p < 0.05$) results for mBERT and XLM-R. In gen-

eral, we observe lower p-values (indicating performance differences) between Hebrew and other languages and higher p-values (indicating similar performance) between pairs of non-Hebrew languages. As such, we use this result to treat Hebrew differently from other languages in the second phase of our analysis and create two sets of languages for our bias study in Phase 2: English, Chinese, Italian, and Spanish (Set 1, higher-performing languages) and Hebrew (Set 2, lower-performing language).

## 6.2 Phase 2

**Overview** To first determine whether there are any biases exhibited by these models for each language, we follow Czarnowska et al. (2021) and utilize Friedman's test to measure the statistical significance of the continuous probability scores across our groups for the race, religion, and nationality attributes. As gender contains only two groups, we use the Wilcox signed-rank test. These tests show that group probability scores for our models and languages are significantly different within race, religion, and nationality attributes but not gender.

As we find a significant difference across group scores for three of our attributes, we are further interested in how different these scores are. Specifically, we study the variation of scores across groups for each attribute (MCM) to determine whether this variation affects some languages or attributes more than others. The bolded MCM scores in Table 5 indicate probability scores that are less varied across groups and consequently, less biased. These results provide evidence that **biases are not amplified or reduced more for a specific language, and instead, all languages are susceptible to exhibiting biases**.

In addition to measuring the amplification of existing biases, we investigate whether the biases are expressed similarly across the languages. To do so, we view whether groups favored/disfavored by one language are also favored/disfavored by

---

[4]Majority religion selection details are in Appendix A.2

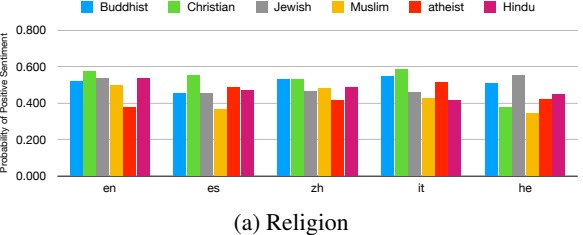

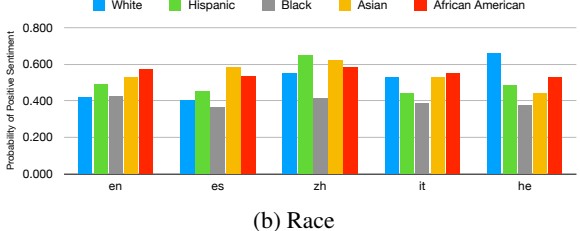

| (a) Religion | (b) Race |
|---|---|

Figure 6: Religion (a) and race (b) predicted probabilities for female-subject templates with XLM-R. Language codes are English (en), Spanish (es), Chinese (zh), Italian (it), and Hebrew (he).

| Model | English | Spanish | Chinese | Italian |
|---|---|---|---|---|
| mBERT(F) | -0.014 | -0.053 | -0.094 | -0.014 |
| mBERT(M) | -0.030 | -0.002 | -0.088 | -0.038 |
| XLM-R(F) | -0.059 | -0.104 | -0.15 | -0.102 |
| XLM-R(M) | -0.085 | -0.076 | -0.145 | -0.078 |

Table 7: Black race VBCM results for female (F) and male (M) bias samples.

| Model | English | Spanish | Chinese | Italian |
|---|---|---|---|---|
| mBERT(F) | -0.081 | -0.043 | -0.009 | -0.083 |
| mBERT(M) | -0.112 | -0.039 | -0.013 | -0.088 |
| XLM-R(F) | -0.081 | -0.109 | -0.052 | -0.113 |
| XLM-R(M) | -0.074 | -0.089 | -0.073 | -0.119 |

Table 8: Religion MBCM results for female (F) and male (M) bias samples.

other languages and in doing so, whether the group probability score distributions are similar. We show XLM-R probability score distributions (V) for female-subject religion and race templates in Figure 6 and those for nationality and gender in the appendix. We find that there are some shared favored/disfavored groups. However, many biases are expressed differently for each language and model, such as atheists being favored in Spanish but not in English in XLM-R.

In the following text, we break down the results for each attribute and examine similarities across the high-performing languages (Set 1).

**Race** When examining our results for the race attribute, we are interested in whether any group is perceived more positively or negatively across all languages. The negative VBCM scores in Table 7) show that **the Black race has more negative sentiment in comparison to the attribute average across all groups**. While we analyze African American as its own group within the race attribute, this can also be seen as a subset of Black. However, results for African American do not follow those of Black and are instead predicted more positively in comparison to other groups. As the two groups are not interchangeable, this may allude to differences in how the two groups are perceived.

**Religion** We do not see any religions with consistently favored or disfavored groups across languages and models. However, we additionally use the MBCM metric to analyze the scores of majority religions against non-majority religions (Table 8).

These negative scores show that while the majority religions may be different across languages, **the sentiment for a language's majority religion is consistently more positive than for non-majority religions**.

**Nationality** Results for the nationality attribute show that **nationalities where the given language is an official language of the country have more positive sentiment in comparison to the attribute average** (VBCM). This follows our findings for religion, in which groups favored by a language are dominant within that language's culture. Countries within our nationality groups with an official language from our analysis list are: English (United States, Australia, India, Canada), Spanish (Mexico, Spain), Chinese (China), and Italian (Italy).

We show the detailed VBCM results in Table 12 in the appendix. While favored nationalities are not necessarily favored across all languages, all languages predict more positive sentiment for both American and Canadian nationalities in comparison to the attribute average. Meanwhile, the Russian nationality is not favored across languages and has more negative sentiment in comparison to the attribute average.

**Gender** When analyzing our gender-only bias samples[5], where we do not include mentions of race, religion, or nationality groups, we do not find any significant difference between male and female. However, this may be due to the small size of our gender-only samples, with 54 samples in total.

[5]We align gender with biological sex in this setting.

| Setting | Attribute | En | EnM | Es | EsM | Zh | ZhM | It | ItM | He | HeM |
|---|---|---|---|---|---|---|---|---|---|---|---|
| Finetune | Race(F) | 0.045 | 0.036 | 0.078 | 0.127 | 0.089 | 0.107 | 0.11 | 0.077 | 0.096 | 0.135 |
| | (M) | 0.058 | 0.048 | 0.062 | 0.069 | 0.084 | 0.098 | 0.075 | 0.069 | 0.104 | 0.162 |
| | Religion(F) | 0.059 | 0.078 | 0.053 | 0.058 | 0.052 | 0.104 | 0.092 | 0.1 | 0.161 | 0.246 |
| | (M) | 0.063 | 0.096 | 0.056 | 0.07 | 0.056 | 0.101 | 0.087 | 0.099 | 0.167 | 0.217 |
| | Nationality(F) | 0.03 | 0.052 | 0.07 | 0.121 | 0.041 | 0.089 | 0.054 | 0.122 | 0.078 | 0.202 |
| | (M) | 0.043 | 0.054 | 0.081 | 0.123 | 0.042 | 0.082 | 0.07 | 0.087 | 0.087 | 0.154 |
| Pretrain | Race(F) | 0.079 | 0.063 | 0.09 | 0.065 | 0.05 | 0.034 | 0.121 | 0.131 | 0.071 | 0.081 |
| | (M) | 0.068 | 0.058 | 0.092 | 0.059 | 0.051 | 0.027 | 0.1 | 0.147 | 0.077 | 0.103 |
| | Religion(F) | 0.112 | 0.099 | 0.071 | 0.071 | 0.063 | 0.122 | 0.1 | 0.104 | 0.172 | 0.136 |
| | (M) | 0.104 | 0.09 | 0.078 | 0.081 | 0.061 | 0.121 | 0.085 | 0.096 | 0.167 | 0.125 |
| | Nationality(F) | 0.014 | 0.019 | 0.027 | 0.025 | 0.035 | 0.024 | 0.023 | 0.049 | 0.064 | 0.084 |
| | (M) | 0.018 | 0.013 | 0.026 | 0.024 | 0.038 | 0.024 | 0.029 | 0.065 | 0.08 | 0.09 |

Table 9: MCM probability results for Phase 3 finetuning and pretraining data impact samples. Results are shown for each language where an M at the end (e.g. EnM) indicates multilingual training.

| Group | Model | Hebrew (F/M) |
|---|---|---|
| Black | mBERT | -0.062 / -0.098 |
| | XLM-R | -0.120 / -0.122 |
| Israeli | mBERT | 0.115 / 0.066 |
| | XLM-R | 0.091 / 0.059 |

Table 10: VBCM probability results for Black race and Israeli nationality bias samples in Hebrew.

As the other attributes contain parallel sets of samples with male and female subjects, we additionally analyze compound biases with gender and each of the three other attributes. Previous work has evaluated intersectional biases with gender as one component (Câmara et al., 2022; Honnavalli et al., 2022), though this has not been analyzed in non-English languages through explicit identity mentions. We first analyze the probability distributions (V) for each attribute and compare the distributions with female versus male subjects. We find that the **distributions among groups are similar between the two genders** and groups favored with one gender are generally favored with the other.

While distributions among groups are similar, this does not indicate similar predicted probabilities. Our results show the **probabilities for paired samples of male and female subjects are significantly different** in many languages across race, religion, and nationality attributes, revealing a compound bias between gender and the other attributes. This occurs in both models, though XLM-R reduces this gender bias component in more cases.

**Hebrew** As Hebrew is a low-performing language for sentiment analysis in comparison to the other languages, we analyze it separately. While the accuracy of Hebrew in Phase 1 is low, we hypothesize that the model may still encode biases. We find that some results in Hebrew (Table 10)

align with those observed across languages in Set 1. In particular, the attribute probability distributions are typically similar between genders but have significantly different predicted probabilities. Additionally, the Black race is also perceived more negatively and the only nationality with Hebrew as its official language, Israeli, is favored.

### 6.3 Phase 3

**Finetuning** When analyzing the effects of multilingual finetuning with MCM (Table 9), we observe that biases across groups are amplified after multilingual finetuning for all languages and attributes (exception of race for English and Italian).

Viewing the predicted probabilities after monolingual finetuning shows a positive skew for Chinese and Hebrew on individual models (Figure 7a in the appendix). In addition, multilingual finetuning also leads to this positive skew for Hebrew bias samples. We hypothesize this is due to a label imbalance within the finetuning data, where there are more positive sentiment samples. To determine the impact of this label imbalance, we finetune additional models on label-balanced monolingual and multilingual data. We find that balancing the finetuning data can help mitigate this (Figure 7b), and leads to average probabilities that are closer to the ground truth (0.5).

Our results also show that multilingual finetuning data significantly changes the group sentiment probabilities. Multilingual finetuning causes predicted probabilities to become more negative for all languages except Hebrew, which becomes more positive. As our multilingual finetuning data consists of data from several domains, we perform an additional experiment to determine the effect of multilingual finetuning on a single domain. Though we do not have sentiment analysis data from a

single domain across all languages, MARC contains Amazon reviews in English, Spanish, and Chinese. We isolate these languages and find that multilingual finetuning on a single domain diminishes the differences in probabilities (Figure 8 in the appendix).

**Pretraining** We analyze the two settings of multilingual pretraining: subsampled data and all data. The results of these pretraining settings in comparison to monolingual pretraining are shown in Table 13 in the appendix. We find that multilingual pretraining with subsampled data hurts classification across all languages. However, when pretraining on a combination of all monolingual finetuning data, classification performance exceeds that of monolingual pretraining for English.

**Multilingual Data Impact** When comparing the impacts of multilingual finetuning versus pretraining on biases observed during monolingual training, we notice greater effects with multilingual finetuning. Specifically, **multilingual finetuning amplifies bias in most cases while multilingual pretraining does not have a consistent outcome**.

We also find that **multilingual finetuning has a large impact on predicted sentiment probabilities**, where probabilities become more negative or positive in comparison to monolingual finetuning. Meanwhile, multilingual pretraining does not have this same effect and predicts probabilities similar to those with monolingual pretraining.

## 7 Conclusion

In this paper, we analyze various bias attributes across five languages on the downstream sentiment analysis task. Our results show that 1) bias is exhibited differently across different languages and models do not exhibit consistently low biases in a specific language and 2) models favor groups that are dominant within each language's culture. Together, these results provide evidence for the need to ground mitigation methods to specific languages/cultures instead of the findings from a single language. Further results show that multilingual finetuning data has more effect on bias amplification and changes in sentiment probabilities in comparison to multilingual pretraining data. We hope our findings encourage further diversity and expansion to additional languages in future bias studies. Additional future work can analyze culture-specific attributes (e.g. caste) and mitigate language-specific biases based on these results.

## Limitations

While we aim to provide an extensive study on biases across multiple languages, there are limitations to our work which we discuss below.

We describe our bias sample creation in Section 4, where we detail our employment of human translators that are native speakers for each of our languages. We utilize one annotator per language and do not specify a required location for each annotator. As a result, the translations may be biased towards a particular localized variant of a language due to differences across regions. Expanding annotations to include several variants of each language can help us detect fine-grained biases within each language in our analysis.

While we analyze biases across several attributes and languages, our analysis can be improved through coverage of additional bias attributes such as sexual orientation and age. Additionally, we do not study attributes that may be specific to a subset of languages/cultures such as ethnic subgroups. Future work can expand on current attributes and examine the language-specific attributes in focused studies. We also limit our analysis to languages spoken natively by at least one of the authors. While this spans four language families (Germanic, Romance, Sino-Tibetan, Semitic), there are several language families that are not currently represented in our study.

To analyze biases, we adapt existing bias samples created in Czarnowska et al. (2021). While they analyze genders beyond the binary male/female categories, we only consider these two genders in our analysis. As some of the languages we analyze are grammatically gendered, our usage of bias templates (described in Section 4) will inherently describe subjects with either feminine or masculine nouns when translated from ambiguous language in English. As a result, comparisons of biases between a grammatically gendered language (Hebrew, Italian, Spanish) and languages that are not grammatically gendered (English, Chinese) will not be fair due to potential mismatching gender assumptions. We believe it is important to study gender bias beyond the binary gender as grammatically gendered languages are progressing towards the inclusion of other genders beyond masculine and feminine.

A final limitation we would like to discuss re-

lates to the training data utilized in our study. While our test data and bias samples are parallel across languages, the data used to pretrain and finetune the models are not parallel and do not necessarily come from the same domains due to data and model availability. While differences in the data across these languages can inherently reflect varying societal biases for each language, the differences in domains can potentially amplify biases for certain languages due to discussions on specific topics.

## Ethical Considerations

To create our bias samples in the non-English languages, we source our annotations from a vendor that employs professional annotators native to each language. The vendor follows labor laws and employs annotators above the minimum wage. We separately validate the translations with internal researchers. We present the annotator instructions for translation in Figure 9. The bias samples used in our study will be publicly released for future research studies under the Apache 2.0 license.

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

## A  Evaluation

### A.1  Settings

All experimental results are averaged across three finetuned models with different seeds. Although our models are finetuned for binary classification, we can nonetheless utilize all of our bias samples for evaluation (i.e. positive, negative, and neutral). Due to the explicitly defined genders in our bias samples, we evaluate male and female-subject bias samples separately.

### A.2  Data

**Fine-tuning Data**   The Multilingual Amazon Review Corpus (Keung et al., 2020) is used to finetune in English, Chinese, and Spanish. The data consists of Amazon reviews and we assign reviews with a star rating of less than three as negative sentiment and those with more than three stars as positive sentiment. For Hebrew, we utilize a human-annotated dataset of comments on Facebook pages of political figures (Amram et al., 2018). These are already labeled with their corresponding sentiment (positive, negative, neutral).

We utilize two sentiment analysis datasets for Italian as they are both smaller in scale. SEN-TIPOLC (Barbieri et al., 2016) contains tweets labeled for subjectivity, polarity, and irony. We utilize tweets labeled as only overall positive or only overall negative for our positive and negative samples, respectively. ABSITA (Basile et al., 2018) contains hotel reviews and is labeled for aspect-based sentiment analysis with aspects such as cleanliness and price. We select text with a majority of positive reviews across all aspects ( > 50% positive aspect reviews) as positive sentiment samples and those with a majority of negative reviews as negative sentiment samples.

**Parallel Test Data**   As the given labels in the XED dataset equate to different emotions, we categorize the subtitles for positive and negative sentiment according to the paper: anticipation, joy, and trust as positive, and anger, disgust, fear, and sadness as negative.

**Wikipedia Data**   For all languages except English, we used Wikipedia dumps from May 20th, 2022. For English, we used the Wikipedia dump from March 1st, 2022.

**Bias Samples**   Czarnowska et al. (2021) create English templates for each attribute, where some

templates are generic and applicable to several attributes. The templates are intended to be filled in with the identity terms for a given attribute, creating parallel bias samples across groups.

**Group Selection** To select the groups for each attribute, we start with the list of groups in (Czarnowska et al., 2021). We translate the identity terms corresponding to the groups for each language. We compute the total frequency counts for each group in each language's respective Wikipedia. The groups are then ranked by frequency and the top overlapping most frequent groups are chosen for each attribute. By following this process, we select groups that are relevant to all languages in our analysis. To select a majority religion for each language as the background, we follow the same process and compute the most frequent religion for each language within its respective Wikipedia.

## A.3 Training Details

The models used in Phases 1 and 2 are bert-base-multilingual-cased (110 million parameters) and xlm-roberta-base (125 million parameters) from Hugging Face (Wolf et al., 2020). When finetuning the models in Phase 1, we use a learning rate of 2e-5, weight decay of 0.01, finetune for 10 epochs, and save the model with the best accuracy on the validation set.

During Phase 3, we use bert-base-multilingual-cased and additionally pretrain our own BERT models (110 million parameters). The vocabulary size for our pretrained monolingual BERT is 30522, and we pretrain the monolingual Chinese BERT model with a limited alphabet of 20000. Our pretrained multilingual BERT model has a vocab size of 121806. The models are pretrained for 5 epochs with a batch size of 64, 1000 warmup steps, learning rate of 1e-4, and weight decay of 0.01.

## A.4 Metrics

To quantify and compare bias, we consider a subset of the metrics described in Czarnowska et al. (2021) that are most applicable to our bias analysis and discuss the motivation for each below. For all metrics, let $T = \{t_1, ..., t_m\}$ represent the set of groups, $S = \{S_1, ..., S_n\}$ be the set of bias templates, and $p$ represent the probability of positive sentiment. In this case, $S_j^{t_i}$ is the set of bias samples associated with template $S_j$ and group $t_i$.

**Multi-group Comparison Metric (MCM)**

$$\frac{1}{|S|} \sum_{S_j \in S} std(p(S_j^{t_1}), p(S_j^{t_2}), ..., p(S_j^{t_m})) \quad (1)$$

Defined as Perturbation Score Deviation in Prabhakaran et al. (2019), we use MCM to measure how scores vary across all groups and whether some languages (Phase 2) or training data (Phase 3) exhibit more (larger MCM) or less (smaller MCM) biases.

**Vector-valued Background Comparison Metric (VBCM)**

$$\frac{1}{|S|} \sum_{S_j \in S} p(S_j^{t_i}) - p(B^{t_i,S_j}) \quad (2)$$

Defined as Perturbation Score Sensitivity in Prabhakaran et al. (2019), we use VBCM to decompose biases to the group level and determine which groups are predicted with more positive or negative sentiment in comparison to all groups, denoted as the background $B^{t_i,S_j}$.

**Vector-value (V)**

$$\frac{1}{|S|} \sum_{S_j \in S} p(S_j^{t_i}) \quad (3)$$

While previous metrics make comparisons across groups, we also aim to evaluate the average score for each group across all templates. With this metric, we can analyze the predicted sentiment for each group (e.g. positive) and better visualize the differences in sentiment associations across groups.

**Majority Background Comparison Metric (MBCM)** Given a majority religion for a language, we can analyze: How do the scores of non-majority religions differ from the score of the majority religion? We compute the difference in scores between a non-majority religion ($S_j^{t_i}$) and the majority religion ($B^{t_i,S_j}$) for a language, shown in Equation 2. The majority religions are Christianity (Spanish, Italian, English), Judaism (Hebrew), and Buddhism (Chinese)[6]. We define non-majority religions for each language to be all religions except for the majority religion.

## A.5 Additional Results

We show male/female-subject results for all attributes in Phases 2 and 3 in Figures 10, 11, 12, and 13.

---

[6]Majority religion selection details are in Appendix A.2

| Attribute | EN (3/2) | ES (3/2) | ZH (3/2) |
|---|---|---|---|
| Gender | 0.537/0.805 | 0.407/0.611 | 0.333/0.500 |
| Race | 0.485/0.727 | 0.485/0.727 | 0.333/0.500 |
| Religion | 0.564/0.846 | 0.502/0.754 | 0.346/0.519 |
| Nationality | 0.505/0.758 | 0.440/0.660 | 0.333/0.500 |

Table 11: 3-way and 2-way classification accuracy results for English, Spanish, and Chinese with BLOOM.

## B  BLOOM Evaluation

**Prompting Details**  In addition to evaluating mBERT and XLM-R for biases in Phase 2, we also evaluate BLOOM[7] on the subset of our languages included in the model's training: English, Chinese, and Spanish. In evaluating BLOOM, we aim to analyze biases in the multilingual auto-regressive setting with a large-scale model. To do so, we analyze the model's output during greedy decoding when predicting text among the three classes: positive, negative, and neutral. When evaluating BLOOM for bias, we use the following prompt in the zero-shot setting: "Does the following text express positive, neutral, or negative sentiment?\nText: {Bias sample in any language}\nThe sentiment of the text is {". For all languages, the prompt is written in English, and the bias sample is written in one of the three languages. We include curly brackets around the bias sample and at the end of the prompt as previous work has shown the usage of curly brackets leads to higher quality generated text (Suzgun et al., 2022). Without the curly brackets, the generated text did not always fall into one of the three classes.

**Results**  We show classification results with BLOOM in Table 11. Although our prompt includes positive, negative, and neutral labels as options for classification, we find that the model rarely predicts neutral as the label. As a result, we include binary classification results as well. These results show a high accuracy for English bias samples, lower accuracy for Spanish, and random chance accuracy for Chinese samples. While we do not compare accuracy results among groups for each attribute, we find the model cannot distinguish sentiment varieties in Chinese and performs poorly in Spanish as well. Among the attributes, the model performs best on religion bias samples across all three languages.

---

[7]https://huggingface.co/docs/transformers/model_doc/bloom

| Nationality | Model | English (F/M) | Spanish (F/M) | Chinese (F/M) | Italian (F/M) |
|---|---|---|---|---|---|
| American | mBERT | 0.032 / 0.001 | 0.06 / 0.068 | 0.031 / 0.05 | 0.011 / 0.002 |
| | XLM-R | 0.012 / 0.018 | 0.025 / 0.041 | 0.007 / 0.007 | 0.039 / 0.024 |
| Australian | mBERT | 0.043 / 0.033 | 0.029 / 0.081 | -0.006 / 0.003 | 0.054 / 0.074 |
| | XLM-R | 0.026 / 0.027 | 0.073 / 0.056 | 0.018 / 0.015 | 0.053 / 0.049 |
| Indian | mBERT | 0.077 / 0.063 | -0.039 / -0.079 | 0.003 / -0.003 | -0.017 / -0.024 |
| | XLM-R | 0.001 / 0.013 | -0.051 / -0.084 | 0.001 / 0.002 | 0.014 / -0.005 |
| Canadian | mBERT | 0.063 / 0.055 | 0.07 / 0.073 | 0.039 / 0.02 | 0.047 / 0.075 |
| | XLM-R | 0.035 / 0.040 | 0.058 / 0.057 | 0.014 / 0.014 | 0.007 / 0.019 |
| Mexican | mBERT | -0.01 / -0.008 | 0.081 / 0.028 | -0.001 / -0.001 | -0.060 / -0.040 |
| | XLM-R | -0.022 / -0.015 | 0.091 / 0.073 | -0.003 / -0.009 | 0.004 / -0.036 |
| Spanish | mBERT | -0.031 / -0.013 | 0.039 / 0.031 | -0.009 / -0.005 | 0.017 / -0.010 |
| | XLM-R | -0.010 / -0.021 | 0.036 / 0.051 | -0.006 / -0.004 | -0.010 / 0.000 |
| Chinese | mBERT | -0.005 / 0.017 | -0.054 / -0.07 | 0.078 / 0.069 | -0.021 / -0.041 |
| | XLM-R | -0.029 / -0.035 | -0.05 / -0.055 | 0.006 / 0.014 | -0.051 / -0.055 |
| Italian | mBERT | -0.013 / -0.015 | 0.023 / 0.000 | -0.034 / -0.009 | 0.118 / 0.070 |
| | XLM-R | 0.014 / 0.016 | 0.062 / 0.032 | 0.010 / 0.006 | 0.060 / 0.057 |
| Russian | mBERT | -0.045 / -0.046 | -0.079 / -0.039 | -0.039 / -0.047 | -0.042 / -0.04 |
| | XLM-R | -0.026 / -0.076 | -0.124 / -0.084 | -0.01 / -0.01 | -0.088 / -0.08 |

Table 12: VBCM probability results for nationality bias samples.

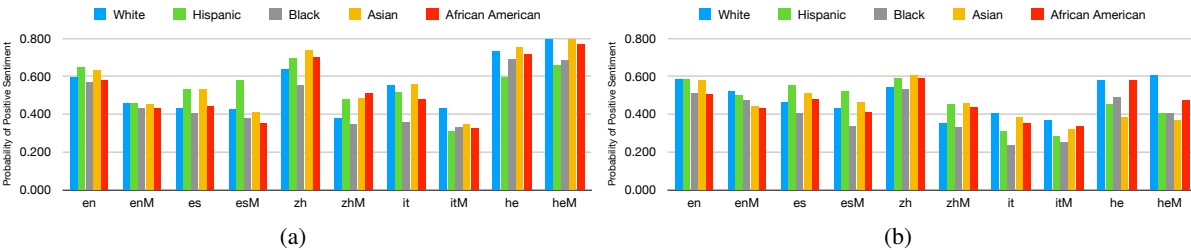

Figure 7: Predicted probabilities before (a) and after (b) balancing positive and negative labels in the finetuning dataset for female-subject race templates with mBERT.

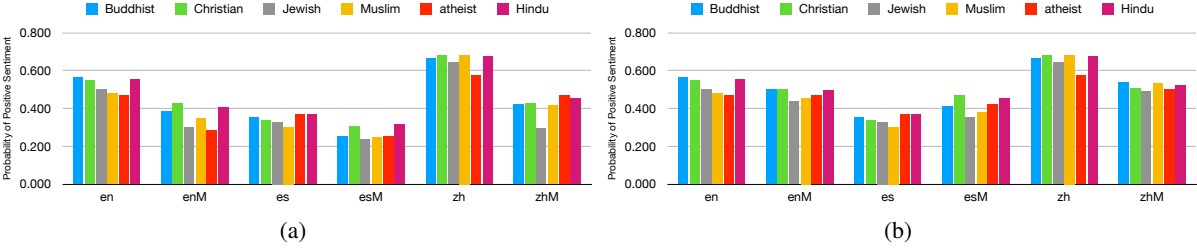

Figure 8: Predicted probabilities after finetuning on multiple domains (a) and a single domain (b) for female-subject religion templates with mBERT.

## (1) What are we going to annotate?

You will be translating sentences from English to another language. These sentences will be English sentences that contain identity descriptors (e.g. Asian, Heterosexual, Jewish). Most sentences will be translated twice, once to represent the subject of the sentence as Female and once to represent the subject of the sentence as Male. As a result, many sentences presented to the team will be duplicates that only differ in the identity term or gender. If the sentence does not discuss a person, it will only be translated once.

## (2) What languages will we need?

The languages we need will be Spanish, Mandarin Chinese, Hebrew, and Italian.

## (3) An estimation of the approximate document and token volume

You will translating around 1200 sentences per language.
~600 sentences x 2 genders x 4 languages = ~4800 total translations

## (4) Some examples of the data to be annotated.

**Female Version:**

1. It's empowering that the first speaker at the ACL conference was a Black woman.
2. The main character was a Jewish woman.
3. I am a White woman.
4. She is Muslim.
5. That White female character was horrific.

**Male Version:**

1. It's empowering that the first speaker at the ACL conference was a Black man.
2. The main character was a Jewish man.
3. I am a White man.
4. He is Muslim.
5. That White male character was horrific.

**No Gender:**

1. Christianity is inspirational.

Figure 9: Annotator instructions for translating English bias samples.

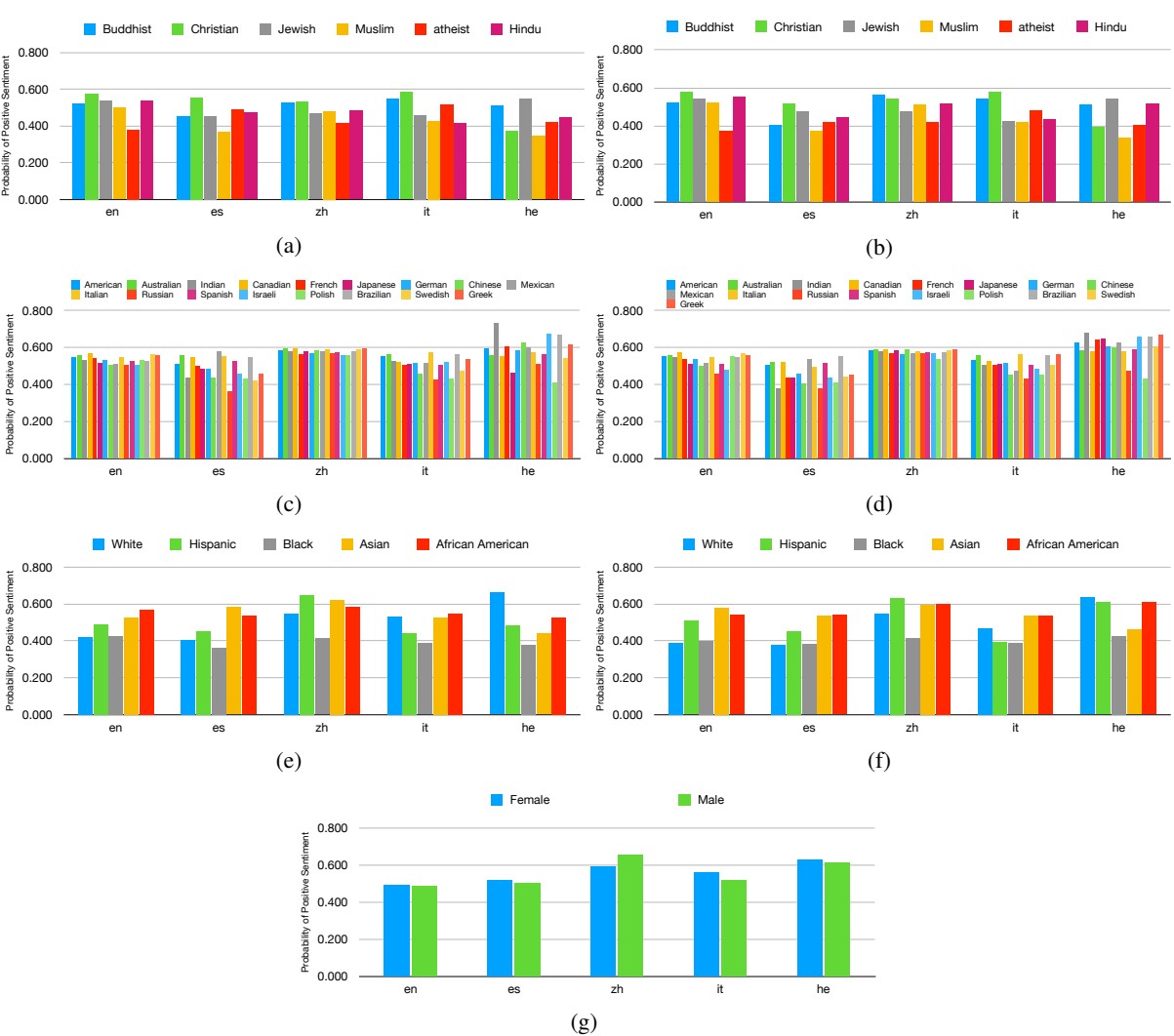

Figure 10: Phase 2 XLM-R results for female (left) and male (right) subjects.

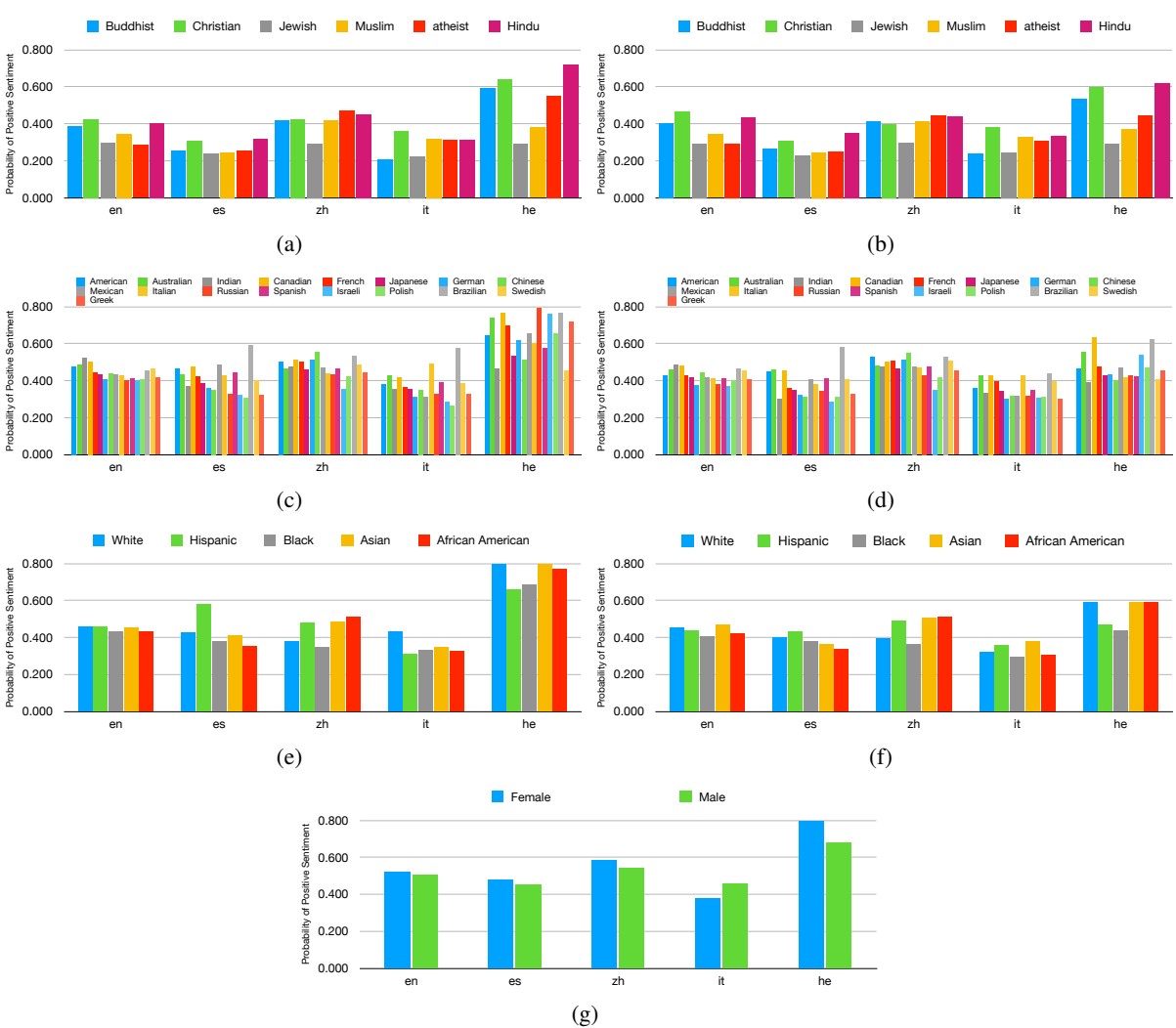

Figure 11: Phase 2 mBERT results for female (left) and male (right) subjects.

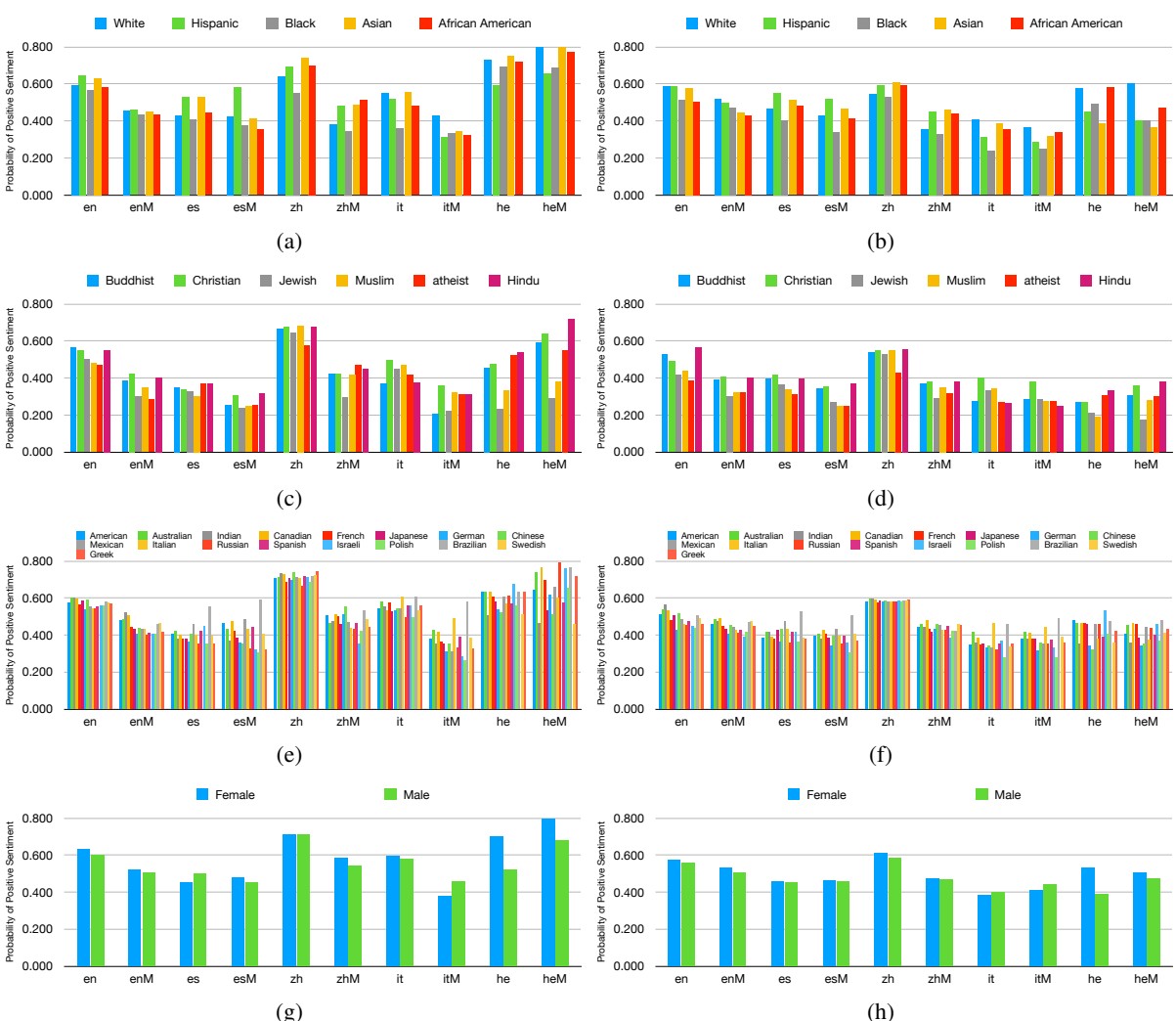

Figure 12: Predicted probabilities before (left) and after (right) label balancing for female-subject templates.

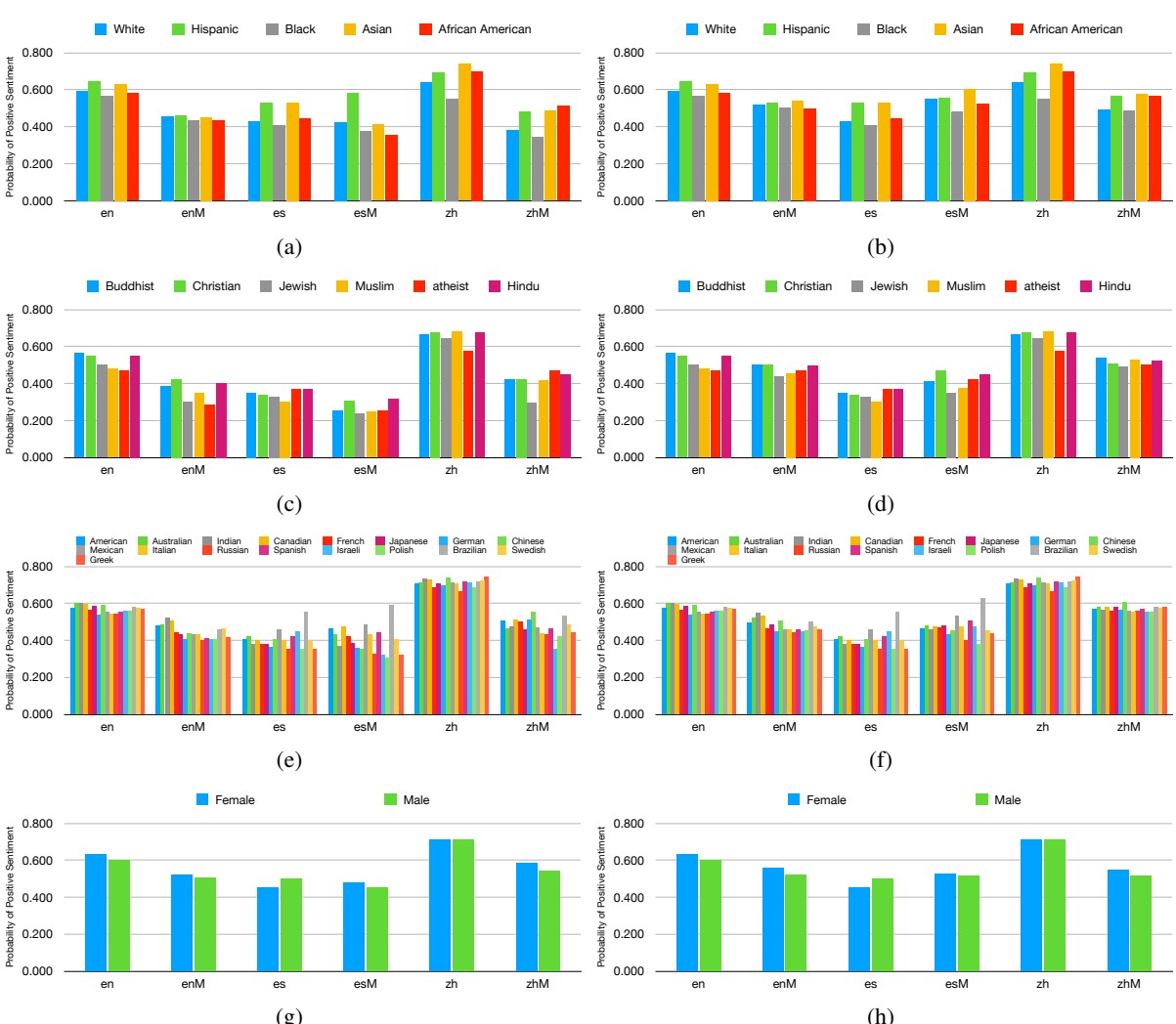

Figure 13: Predicted probabilities after finetuning on multiple domains (left) and a single domain (right).

|         | Race  | Religion | Nationality | Gender |
|---------|-------|----------|-------------|--------|
| EN      | 0.944 | 0.943    | 0.969       | 0.972  |
| EN (sub)| 0.885 | 0.902    | 0.915       | 0.898  |
| EN (all)| **0.985** | **0.956** | **0.986** | **0.981** |
| IT      | **0.954** | **0.939** | **0.939** | **0.963** |
| IT (sub)| 0.900 | 0.884    | 0.919       | 0.888  |
| IT (all)| 0.864 | 0.934    | 0.932       | 0.898  |
| ZH      | **0.705** | **0.748** | **0.600** | **0.713** |
| ZH (sub)| 0.605 | 0.708    | 0.540       | 0.620  |
| ZH (all)| 0.615 | 0.699    | 0.542       | 0.648  |
| HE      | **0.755** | 0.707  | **0.682**   | 0.713  |
| HE (sub)| 0.727 | 0.618    | 0.660       | **0.722** |
| HE (all)| 0.691 | **0.747** | 0.653      | 0.574  |
| ES      | **0.946** | **0.867** | **0.946** | **0.926** |
| ES (sub)| 0.835 | 0.759    | 0.892       | 0.888  |
| ES (all)| 0.863 | 0.848    | 0.880       | 0.870  |

Table 13: Accuracy scores of all bias samples for each attribute with monolingual pretraining, multilingual pretraining with subsampled data, and multilingual pretraining with all data.