# OpenReview forum: "Comparing Biases and the Impact of Multilingual Training across Multiple Languages"
_EMNLP/2023/Conference — EMNLP 2023 Main_

### Official Review · Reviewer_WYR1 · 2023-07-30

**Soundness:** 5

**Excitement:**

4: Strong: This paper deepens the understanding of some phenomenon or lowers the barriers to an existing research direction.

**Paper Topic And Main Contributions:**

The paper presents a multilingual study of race, religion, nationality, and gender biases across six different languages in monolingual and multilingual encoder models. The study is performed using novel metrics for extrinsic bias evaluation with sentiment analysis as a downstream task. In particular, it was interesting to see how models tend to associate positive sentiment with race/nationalities and the religion(s) that relate to a specific language.

**Reasons To Accept:**

The paper provides a new multilingual dataset that could be helpful for future studies on biases in monolingual and multilingual models.

The experiments performed by the authors are extensive and insightful, with multiple BERT models pre-trained from scratch to analyze the impact of pre-training in monolingual and multilingual settings, which little work in the literature has performed before. My suggestion to the authors would be making those models available on HuggingFace for better reproducibility if possible.


**Reasons To Reject:**

The following are a few concerns about the paper:

It is unclear from the paper how the adopted metrics should be interpreted. As currently presented, it is easy to spot how bias varies across different languages, but difficult to understand the extent of bias for a particular language/attribute. Further, the presentation of the results  as a table (in Table 9) was not very helpful to spot trends if existing. My suggestions would be to provide more clarification on the interpretability of the metrics/results and modify the presentation of Table for better clarity.

One particular weakness is the lack of analysis of the reason behind amplification of bias during fine-tuning, but not during pre-training. I believe adding such an analysis or discussion would make the paper stronger.


**Reproducibility:**

4: Could mostly reproduce the results, but there may be some variation because of sample variance or minor variations in their interpretation of the protocol or method.

**Reviewer Confidence:**

5: Positive that my evaluation is correct. I read the paper very carefully and I am very familiar with related work.

---

> ### Author Rebuttal · Authors · 2023-08-28
>
> Thank you for your comments! We hope our responses below address any issues that have been raised:
>
> - For the adopted metrics, we will provide more in-depth details of each metric in the revision. The MCM metric is used to measure the variance of probability scores across groups. A smaller MCM score means that probability scores are less varied across groups and therefore less biased (seen as the bolded scores in Table 6). VBCM measures each group’s score against the average score of all groups. This is intended to help us analyze whether a group is consistently more negative (seen as negative scores in Table 7) or positive in comparison to the overall average. MBCM is similar to VBCM but instead of comparing against the group average, we only compare a group’s score to the majority group’s score. This metric allows us to determine whether non-majority groups are consistently more positive or more negative (seen as negative scores in Table 8) in comparison to majority religions for each language (e.g. in English we can compare each group’s score to Christian’s score). We include more details on each of our chosen metrics in the appendix. Regarding Table 9, we will modify this to better visualize the results of the experiment in our revision.
>
> - With respect to our current results in Phase 3, one hypothesis behind the amplification of bias during finetuning relates to the similarity of the pretraining data across languages (pretraining data stems from Wikipedia) while the domains of finetuning data vary across our languages. We acknowledge this requires further analysis and can be considered a potential direction for future work.

---

### Official Review · Reviewer_jH5Q · 2023-08-04

**Soundness:** 4

**Excitement:**

3: Ambivalent: It has merits (e.g., it reports state-of-the-art results, the idea is nice), but there are key weaknesses (e.g., it describes incremental work), and it can significantly benefit from another round of revision. However, I won't object to accepting it if my co-reviewers champion it.

**Paper Topic And Main Contributions:**

This paper compares biases across languages in multilingual models, specifically mBERT and XLM-R, and using a sentiment analysis task. They find some preference for the majority groups associated with each language.

**Questions For The Authors:**

- What do you mean by multilingual finetuning on a particular language?
- Why have a whole extra evaluation on BLOOM and not even mention it in the body of the paper?

**Reasons To Accept:**

The question of how and whether biases vary in multilingual models is definitely relevant.

**Reasons To Reject:**

- The MCM, VBCM, MBCM measures are not well explained and make it very hard to interpret the tables and charts using the information in the paper.
- I found it hard to understand some of the experiment descriptions/interpretations, particularly Phase 3.

**Reproducibility:**

4: Could mostly reproduce the results, but there may be some variation because of sample variance or minor variations in their interpretation of the protocol or method.

**Reviewer Confidence:**

3: Pretty sure, but there's a chance I missed something. Although I have a good feel for this area in general, I did not carefully check the paper's details, e.g., the math, experimental design, or novelty.

**Typos Grammar Style And Presentation Improvements:**

Footnote 4 is unnecessary given you’ve already stated the limitation to the binary and in fact “aligning gender with biological sex” is not possible based on pronouns and identity terms.

---

> ### Author Rebuttal · Authors · 2023-08-28
>
> Thank you for your comments! We hope our responses below address any issues that have been raised:
>
> - For the MCM, VBCM, MBCM measurements, we will provide more in-depth details of each metric in the revision. The MCM metric is used to measure the variance of probability scores across groups. A smaller MCM score means that probability scores are less varied across groups and therefore less biased (seen as the bolded scores in Table 6). VBCM measures each group’s score against the average score of all groups. This is intended to help us analyze whether a group is consistently more negative (seen as negative scores in Table 7) or positive in comparison to the overall average. MBCM is similar to VBCM but instead of comparing against the group average, we only compare a group’s score to the majority group’s score. This metric allows us to determine whether non-majority groups are consistently more positive or more negative (seen as negative scores in Table 8) in comparison to majority religions for each language (e.g. in English we can compare each group’s score to Christian’s score). We include more details on each of our chosen metrics in the appendix.
>
> - Thank you for the feedback on Phase 3. As a summary, we evaluate the impacts of multilingual pretraining and finetuning separately. To analyze multilingual pretraining, we pretrain individual monolingual models and two multilingual ones (one with sampled data to match our monolingual data size and one with all the monolingual pretraining data combined). To study each of our five languages, we finetune the monolingual model and multilingual model on only that language and compare the results.
> When analyzing the impact of multilingual finetuning, we only use the off-the-shelf multilingual BERT model and compare results for each language when finetuning on data from all of our languages versus monolingual finetuning on only one language.
> In our results, we find that the MCM score increases (i.e. bias is amplified) for our languages when we do multilingual finetuning versus monolingual finetuning. There is no consistent outcome with our analysis on the impact of multilingual pretraining. We will include further details of the experiment descriptions and interpretations in our revision.
>
> - For multilingual finetuning, we finetuned a model on all of our languages (instead of only one) and then observed the behavior on each of the languages. Our understanding of the question on multilingual finetuning is in reference to line 515. In this sentence, we express that monolingual finetuning with Chinese and Hebrew on individual models leads to a positive skew in predicted probabilities for each of those languages’ bias samples. In addition, multilingual finetuning also leads to this positive skew for only Hebrew bias samples. This visualization can be seen in Figure 4a in the appendix. We will rewrite this sentence for more clarity in our revision.
>
> - In regards to the BLOOM evaluation, we focus the main body of the paper on models with comparable results. We include the BLOOM evaluation in the appendix as the model lacks coverage of all the languages we study in the main body of the paper. In particular, BLOOM was not trained on both Italian and Hebrew, therefore limiting our evaluation of it to only English, Chinese, and Spanish. As our paper is focused on comparing biases across languages, we choose to include this additional evaluation in the appendix since we cannot comprehensively perform this full analysis. We will clarify this in our revision.

---

### Official Review · Reviewer_3PjR · 2023-08-05

**Soundness:** 5

**Excitement:**

4: Strong: This paper deepens the understanding of some phenomenon or lowers the barriers to an existing research direction.

**Paper Topic And Main Contributions:**

The paper analyses bias for or against different groups in sentiment prediction across multiple languages. To this end, template sentences are generated, which are then filled with different groups, comparing the results of sentiment prediction. The paper evaluates multiple settings, analysing the influence of pre-training as well as fine-tuning data on the bias present in the resulting model.

**Questions For The Authors:**

a) l. 246: How does the oversampling work?

**Reasons To Accept:**

a) Very well-designed study
The study presented in the paper is very well designed, taking into account almost all potential problems that I could come up with (different amount of training data in different languages, different performance of the classifiers, ...)

b) Comprehensive evaluation

c) Well written paper


**Reasons To Reject:**

a) Minor: Different sentiment training data for different languages
This is the only factor I found that was not controlled for in the study: The training datasets for the different languages are not the same and partially come from different domains (SENTIPOLC contains Twitter data, while most other datasets contain Amazon reviews). This might skew the model's performance - however, since the authors make sure that the performance of the models is similar, this should not be a major problem.

Overall, this is a very well written paper, describing a well-designed study on a relevant topic. I therefore recommend accepting it.

**Reproducibility:**

4: Could mostly reproduce the results, but there may be some variation because of sample variance or minor variations in their interpretation of the protocol or method.

**Reviewer Confidence:**

4: Quite sure. I tried to check the important points carefully. It's unlikely, though conceivable, that I missed something that should affect my ratings.

**Typos Grammar Style And Presentation Improvements:**

- l. 108: "measure[s]"

---

> ### Author Rebuttal · Authors · 2023-08-28
>
> Thank you for your comments! We hope our response below addresses your question:
>
> Question A: We perform oversampling so that the number of articles for low-resource languages matches that of our mid-resource language, Italian (third largest in terms of article sizes for our languages). When we oversample low-resource languages, we randomly sample and duplicate $x$ articles from the existing set of articles for that language. In this case $x = a - b$, where $a$ is the number of articles for Italian, and $b$ is the number of articles for the low-resource language.

---

### Meta-Review · Area_Chair_MEEg · 2023-09-19

**Recommendation:** 4

**Metareview:**

The paper "Comparing Biases and the Impact of Multilingual Training across Multiple Languages" presents work on detecting bias similarities and differences across several language and based on monolingual and multi-lingual training data.

Arguments against the paper raised by the reviewers are related to details in the explanation of several aspects of the paper.
In general, the reviewers are very much in favour of the paper and see the presented work as an important topic and that the paper addresses this paper in a very appropriate way.

---

### Decision · Program_Chairs · 2023-10-07

**Decision:**

Accept-Main

**Comment:**

The paper "Comparing Biases and the Impact of Multilingual Training across Multiple Languages" presents work on detecting bias similarities and differences across several language and based on monolingual and multi-lingual training data.

Arguments against the paper raised by the reviewers are related to details in the explanation of several aspects of the paper.
In general, the reviewers are very much in favour of the paper and see the presented work as an important topic and that the paper addresses this paper in a very appropriate way.